# *Rubus ellipticus* Sm. Fruit Extract Mediated Zinc Oxide Nanoparticles: A Green Approach for Dye Degradation and Biomedical Applications

**DOI:** 10.3390/ma15103470

**Published:** 2022-04-06

**Authors:** Jyoti Dhatwalia, Amita Kumari, Ankush Chauhan, Kumari Mansi, Shabnam Thakur, Reena V. Saini, Ishita Guleria, Sohan Lal, Ashwani Kumar, Khalid Mujasam Batoo, Byung Hyune Choi, Amanda-Lee E. Manicum, Rajesh Kumar

**Affiliations:** 1School of Biological and Environmental Sciences, Faculty of Sciences, Shoolini University of Biotechnology & Management Sciences, Solan 173212, Himachal Pradesh, India; dhatwaliajyoti3096@gmail.com (J.D.); shabnamthakur780@gmail.com (S.T.); ishita.thakur93@gmail.com (I.G.); sohanlal4810@gmail.com (S.L.); 2Chettinad Hospital and Research Institute, Chettinad Academy of Research and Education, Kanchipuram 603103, Tamil Nadu, India; ankushchauhan18@gmail.com; 3Advanced School of Chemical Sciences, Shoolini University of Biotechnology & Management Sciences, Solan 173212, Himachal Pradesh, India; mansi528sharma@gmail.com; 4Central Research Laboratory MMIMSR, Department of Biotechnology MMEC, Maharishi Markandeshwar (Deemed to be University), Mullana 133207, Haryana, India; reenavohra10@gmail.com; 5Patanjali Research Institute, Haridwar 249405, Uttarakhand, India; ashu5157@gmail.com; 6King Abdullah Institute for Nanotechnology, College of Science, King Saud University, Building No. 04, Riyadh 11451, Saudi Arabia; khalid.mujasam@gmail.com; 7Department of Biomedical Sciences, Inha University College of Medicine, 100 Inha-ro, Incheon 22212, Korea; bryan@inha.ac.kr; 8Department of Chemistry, Faculty of Science, Arcadia Campus, Tshwane University of Technology, Pretoria 0183, South Africa; manicumae@tut.ac.za; 9Department of Physics, Faculty of Physical Sciences, Sardar Vallabhbhai Patel Cluster University, Mandi 175001, Himachal Pradesh, India

**Keywords:** ZnO-NPs, photocatalyst, antioxidant, antimicrobial, anticancer activity

## Abstract

*Rubus ellipticus* fruits aqueous extract derived ZnO-nanoparticles (NPs) were synthesized through a green synthesis method. The structural, optical, and morphological properties of ZnO-NPs were investigated using XRD, FTIR, UV-vis spectrophotometer, XPS, FESEM, and TEM. The Rietveld refinement confirmed the phase purity of ZnO-NPs with hexagonal wurtzite crystalline structure and p-63-mc space group with an average crystallite size of 20 nm. XPS revealed the presence of an oxygen chemisorbed species on the surface of ZnO-NPs. In addition, the nanoparticles exhibited significant in vitro antioxidant activity due to the attachment of the hydroxyl group of the phenols on the surface of the nanoparticles. Among all microbial strains, nanoparticles’ maximum antibacterial and antifungal activity in terms of MIC was observed against *Bacillus subtilis* (31.2 µg/mL) and *Rosellinia necatrix* (15.62 µg/mL), respectively. The anticancer activity revealed 52.41% of A549 cells death (IC_50_: 158.1 ± 1.14 µg/mL) at 200 μg/mL concentration of nanoparticles, whereas photocatalytic activity showed about 17.5% degradation of the methylene blue within 60 min, with a final dye degradation efficiency of 72.7%. All these results suggest the medicinal potential of the synthesized ZnO-NPs and therefore can be recommended for use in wastewater treatment and medicinal purposes by pharmacological industries.

## 1. Introduction

Nanotechnology is an important field of modern technology that deals with the study of particles having a size of 1–100 nm [1]. The nanoparticles (NPs), which are synthesized through this technology, have various applications such as detectors, surface coating agents, catalysts, and antimicrobial agents [2]. Different types of metal oxides NPs have been synthesized through this technique like silver oxide (AgO), silicon dioxide (SiO_2_), titanium dioxide (TiO_2_), indium (III) oxide (In_2_O_3_), tin (IV) oxide (SnO_2_), zinc oxide (ZnO) and copper oxide (CuO), etc. After SiOTiO_2_, ZnO is one of the most abundantly produced metal oxides [3]. The Zinc oxide nanoparticles (ZnO-NPs) are easy to synthesize, showed less toxicity, therefore, have been reported as the most popular metal oxide nanoparticle [4]. In addition to these, their roles are reported in the literature on drug delivery, wound healing, bioimaging, antimicrobial, antidiabetic, and anticancer activities [4]. Due to zero toxicity, they are widely utilized in the cosmetic industry (in sunscreen and facial creams) [4,5], and food industries as additives [6].

The various methods that have been used for the synthesis of ZnO-NPs are the physical, chemical, hybrid, microwave, and biological methods [7,8,9,10,11,12]. Among all these methods, the most abundantly used method currently is green synthesis, i.e., plant-mediated synthesis of nanoparticles because of its simplicity, cost-effectiveness, and environmentally friendly behavior [13].

Due to the presence of phytochemicals such as phenols, flavonoids, and terpenoids in the plants, they are used for the synthesis of nanoparticles and act as non-toxic reducing, capping, and stabilizing agents [14]. The role of ZnO-NPs is also reported in various biological applications such as antibacterial, antifungal, and anticancer activities. Additionally, due to their catalytic activity, ZnO-NPs are also used for wastewater treatment [15].

The ZnO-NPs synthesized from the extracts of *Rosa indica*, *Justicia adhatoda*, *Vitex negundo*, *Mentha pulegium*, and *Cassia fistula* were reported to inhibit the growth of broad-spectrum pathogens [16,17,18,19,20]. Currently, plant-mediated synthesized nanoparticles are observed to possess photocatalytic activity [15,20,21,22] and they have shown more effective results than chemically prepared nanoparticles [23]. *Rubus ellipticus* Sm., commonly known as yellow Himalayan raspberry, Aiselu, or Aekhae, is an evergreen shrub belonging to the family Apocynaceae. It is native to China, Indonesia, the Indian subcontinent, and Sri Lanka [24]. Its fruits are rich in glycosides, flavonoids, phenols, resin, pectin, and tannins [25,26]. They are commonly used for the treatment of fever, cough, dysentery, constipation, diarrhea, curing bone fracture, and relieving stomach worms in children [25]. In addition, the fruit extract was also reported to possess good antioxidant, antimicrobial, and anticancer activity [26]. Due to the various biological applications of plant-mediated nanoparticles, the present study focuses on the synthesis of environmentally friendly, non-toxic ZnO-NPs using the fruit extract of *R. ellipticus*. These particles are further screened for antimicrobial, anticancer, and photocatalytic activity.

## 2. Material and Methods

### 2.1. Plant Material Collection

*R. ellipticus* fruits were collected from the Shimla (2000 masl) district, Himachal Pradesh, India. The collected fruits were dried in an oven at 40 °C, crushed using a grinder, and the coarse powder was stored in an airtight container for further use. The authentication of the plants was done in the Botanical Survey of India (BSI), Dehradun, India with accession number 117.

#### Extract Preparation

The fruit extract was prepared by dissolving 10 g of powdered sample in 100 mL of distilled water in a water bath at 60 °C for 8 h. The extract was filtered through Whatman filter paper number 41 and dried in the oven at 37 °C. The dried extract was collected and stored at 5 °C in a refrigerator for further analysis [27].

### 2.2. Synthesis of ZnO-NPs

ZnO-NPs were prepared from the fruit’s aqueous extract of *R. ellipticus* by following the method of Chauhan et al. [27]. For the preparation of ZnO-NPs, the 10 mL of fruits aqueous extract (100 mg of fruits’ aqueous extract in 10 mL of distilled water) was mixed with 50 mL of zinc acetate dihydrate (0.5 M; Loba Chemie, Mumbai, India; 98% purity) in a beaker (Figure 1). To the solution, 5 mL of sodium hydroxide (0.2 M) was added, and the whole solution was magnetically stirred for 20 min. at 40 °C temperature. The whole solution was magnetically stirred for 20 min. at a temperature of 40 °C. The resulting solution in the flask was stirred for two hours until white precipitates formed. The precipitates were collected, centrifuged, and washed with deionized water for the removal of impurities. The sample was collected and then dried at 60 °C for 24 h.

### 2.3. Characterization of ZnO-NPs

The characterization of the synthesized ZnO-NPs was done by Fourier transform infrared spectroscopy (FTIR; Perkin Elmer, Waltham, MA, USA) at the resolution of 1 cm^−1^ and a scan range of 4000 cm^−1^ to 250 cm^−1^) for functional group identification, X-ray photoelectron spectroscopy (XPS; Prevac, Rogów, Poland) for determination of elemental composition, chemical state, and electronic state of the elements in the material, X-ray diffraction spectroscopy (XRD; PANalytical, Almelo, Netherlands) for the determination of amorphous and crystalline nature of the material, scanning electron microscope [(SEM-Mapping; SU8010 Series) (Hitachi, Tokyo, Japan)] for checking texture and elemental composition of nanoparticles, and Transmission electron microscope [TEM; JEM 2100 Plus (JEOL, New Delhi, India)] used for size and morphology of nanoparticles at Sophisticated Analytical Instrumentation Facility (SAIF), Punjab University, Chandigarh, India. The absorption spectrum of green synthesized nanoparticles was analyzed by Ultraviolet-visible spectrophotometry (UV 2450, Shimadzu corporation, Kyoto, Japan) in Advanced Materials Research Centre, Indian Institute of Technology (IIT), Mandi, Himachal Pradesh, India.

### 2.4. Antioxidant Assays

#### 2.4.1. DPPH Free Radical Scavenging Assay

The antioxidant potential of ZnO-NPs synthesized from aqueous fruit extract of *R. ellipticus* was analyzed by following the methodology with slight modifications [28]. In this method, 2,2-diphenyl-1-picrylhydrazyl (DPPH: 0.1 M) and ZnONPs (20–100 µg/mL) were prepared in methanol just before the experiment. Firstly, the 1 mL of each concentration was separately mixed with 1 mL of DPPH into the test tube. After this, the test tubes were incubated in the dark for 30 min. In the last step, a 200 µL reaction mixture was taken from each test tube and poured into a 96-well microtiter plate. The absorbance (Abs) was taken through a multileader (VARIOSKAN-LUX, Thermo Fisher Scientific, Waltham, MA, USA) at 517 nm. This assay was performed in triplicate and repeated for aqueous fruit extract. DPPH was used as a control and ascorbic acid was used as a positive control. Methanol was used as a blank for the assay. The radical scavenging activity was calculated as follows:
$$\mathrm{DPPH}\;\mathrm{activity}\;\left(\%\right)\;=\frac{\left[\mathrm{Control}\;\left(\mathrm{Abs}\right)-\;\mathrm{Test}\;\mathrm{sample}\;\left(\mathrm{Abs}\right)\right]}{\mathrm{Control}\;\left(\mathrm{Abs}\right)}\times 100$$


#### 2.4.2. ABTS Assay

The antioxidant effect of the fruit extract was studied using ABTS (2,2-azino-bis-3-ethyl benzthiazoline-6-sulphonic acid) radical cation decolorization assay [28]. ABTS radical cations (ABTS^+^) were produced by reacting 7 mM ABTS solution with 2.45 mM ammonium persulphate; the mixture was prepared by mixing the two stock solutions in equal quantities and allowed to react for 16 h at room temperature in the dark. After that, 0.1 mL of extract of each sample (at four different concentrations 25 µg/mL, 50 µg/mL, 75 µg/mL, 100 µg/mL) were mixed with 0.9 mL ABTS solution in the test tubes. The reaction mixture was incubated in dark conditions for 30 min. The absorbance was read at 745 nm in a UV-vis spectrophotometer (VARIOSKAN-LUX, Thermo Fisher Scientific, Waltham, MA, USA) and the percentage inhibition was calculated by using the formula:
$$\mathrm{ABTS}\;\mathrm{assay}\;(\%)=\frac{\left[\mathrm{Control}\;\left(\mathrm{Abs}\right)-\;\mathrm{Test}\;\mathrm{sample}\;\left(\mathrm{Abs}\right)\right]}{\mathrm{Control}\;\left(\mathrm{Abs}\right)}\times 100$$


#### 2.4.3. FRAP Free Radical Scavenging Assay

The free radical scavenging activity of ZnO-NPs synthesized from aqueous fruit extract at different concentrations (20–100 µg/mL) was analyzed through ferric ion reducing antioxidant power (FRAP) assay with slight modifications [29]. In this method, 1 mL of each concentration was separately mixed with 1 mL of sodium phosphate buffer (0.2 M in distilled water; pH 6.6) into the test tube. After this, 1 mL of 1% potassium ferricyanide was added into all the test tubes and incubated at 50 °C in a water bath for 20 min. After this, 1 mL of trichloroacetic acid (10%) was added to all test tubes and the mixtures were centrifuged for 10 min. at 5000 rpm. In the next step, 1 mL of the reaction mixture was taken from each sample and 1 mL of distilled water was added to it. After this, 0.2 mL of ferric chloride (0.1% in distilled water) was added to each test tube. In the last step, 200 µL solution was taken from each test tube and poured into the 96-well microtiter plate. The blank was prepared similarly except that potassium ferricyanide (1%) was replaced by distilled water. The absorbance was measured at 700 nm using a multileader (VARIOSKAN-LUX, Thermo Fisher Scientific, Waltham, MA, USA)) [30]. The results were recorded in triplicate and ascorbic acid (20–100 µg/mL) was used as a positive control. The antioxidant capacity of each sample was calculated from the linear calibration curve of ferrous sulfhate (y = 0.004x + 0.0377; R^2^) and expressed as µM FeSO_4_ equivalents.

### 2.5. Antimicrobial Assay

#### 2.5.1. Selection of Strains

For the antibacterial assay, two strains of Gram-positive bacteria [*Staphylococcus aureus* (MTCC 731), *Bacillus subtilis* (MTCC 441)], and Gram-negative bacteria [*Pseudomonas aeruginosa* (MTCC 424), *Escherichia coli* (MTCC 739)] were selected, whereas for antifungal assay two pathogenic strains of fungi [*Fusarium oxysporum* (SR266-9) and *Rosellinia necatrix*; (HG964402.1.)] were selected. The bacterial and fungal strains were obtained from CSIR Institute of Microbial Technology (IMTech, Staines, UK), Chandigarh and School of Microbiology, Shoolini University, Solan, India, respectively.

#### 2.5.2. Antibacterial Activity

##### Disc Diffusion Method

The antibacterial activity of fruits’ aqueous extract and ZnO-NPs were observed by using the disc diffusion method [31]. The bacterial culture (100 µL), with the help of sterile cotton swabs, was uniformly spread on the surface of the nutrient agar plates. A stock solution of ZnO-NPs and fruits aqueous extract was prepared by dissolving 10 mg ZnO-NPs and crude aqueous extract of fruits to 1 mL of dimethyl sulfoxide (DMSO) separately. The 300 µg/mL of ZnO-NPs and fruits aqueous extract were applied separately to each 6 mm sterilized paper disc. Discs were placed into Petri plates with bacterial culture and then placed in an incubator set to 37 °C for 24 h. After 24 h of incubation, an antibiotic zone scale was used to record zones of inhibition data. Each antibacterial assay was performed in triplicate. Dimethyl sulfoxide (solvent) and ampicillin (5 mg/mL) were used as negative and positive controls, respectively.

#### 2.5.3. Antifungal Assay

##### Poison Food Technique

For the antifungal assay, the poison food technique was used with minor modifications [32]. The two fungal strains (*F. oxysporum* and *R. necatrix*) were cultured on potato dextrose agar (PDA) at 25 °C for 7 days before use. The 300 µg/mL of fruits aqueous extract and ZnO-NPs were mixed in 24 mL of PDA per plate separately. After the solidification of the media, a 6 mm diameter of fungal disc was cut with flame sterilized cork borer and then placed at the center of each petri dish. The dish was incubated at 25 °C for seven days. The colony diameter of fungus was measured on the seventh day of incubation. The tests were performed in triplicate. The antifungal activity of the fruit aqueous extract and ZnO-NPs were further compared to the control (dish without fruits aqueous extract and ZnO-NPs). The percentage inhibition of radial mycelial growth over the control was calculated by using the following formula:
$$\mathrm{Inhibition}\;(\%)=\frac{C-T}{C}\times 100$$

where C is the diametric growth of the colony in control, T is the diametric growth in the nanoparticles extract.

#### 2.5.4. Minimum Inhibitory Concentration (MIC)

For the determination of antimicrobial activity, the fruit aqueous extract and ZnO-NPs 96 well microtitre plate method were used [33]. The 12 wells of each row of microtiter plates were filled with 0.1 mL of sterilized NA and PDA broth for bacteria and fungus, respectively. Sequentially, wells 2–11 received an additional 5 mg/mL of plant extract and ZnO-NPs in separate columns. The serial dilution was done by transferring 100 µL of testing samples (500 µg/mL) from the first row of the subsequent wells in the next row of the same column. Finally, a volume of 10 µL was taken from bacterial or fungal suspension and then added to each well to achieve a final concentration of 5 × 10^6^ CFU/mL. The well plates were incubated for 24 h at 37 °C and then 15 µL of resazurin solution, as an indicator, was added to each well. The color change from purple to pink or colorless in the well was then observed visually. The lowest concentration of extract at which color change occurred was recorded as the MIC value.

### 2.6. Anticancer Analysis

#### 2.6.1. Cell Culturing and Maintenance

Cell lines were purchased from the National Centre for Cell Sciences (NCCS), Pune, India, A549 (Human lung adenocarcinoma) to evaluate the anticancer effect of ZnO-NPs by using the 3-(4,5-dimethylthiazol-2-yl)-2,5-diphenyl tetrazolium bromide (MTT) assay. The stock cells were cultured in DMEM (Dulbecco’s Modified Eagle’s Medium) supplemented with inactivated fetal bovine serum (FBS) 10%, penicillin-streptomycin (1%) in a humidified atmosphere of CO_2_ (5%) at 37 °C until confluent. To take out dead cells and debris, the used media and debris were deposed and washed with phosphate-buffered saline (PBS). The trypsin (0.25%) was used for cell dissociation and the pallet was cultivated after the centrifugation process at 3000 rpm for 10 min. For sub-culturing, fresh aliquots were made and transferred to new culture dishes.

#### 2.6.2. In Vitro Anticancer Assay (MTT Assay on A549)

To evaluate the anticancer effect through MTT assay, the method reported by Kumari et al. [34] was used. The lung cancer cells (A549) at a density of 1 × 10^4^ cells/mL were seeded in a 96-well plate and incubated at 37 °C for 24 h under 5% CO_2_. The next day, cancer cells were treated with different concentrations (1.56–200 μg/mL) of ZnO-NPs for 24 h, followed by an MTT assay (4 h incubation) to assess the cell viability. For positive and negative control, Paclitaxel and DMSO were used, respectively. After 4 h, DMSO (100 μL) was added to each well for dissolving the purple-colored complex. Using a microplate reader, the optical density was noted at 595 nm. IC_50_ values of the ZnO-NPs (in triplicate) were further recorded, and cell viability was analyzed using the following formula:
$$\mathrm{Cell}\;\mathrm{cytotoxicity}=\frac{\left(\mathrm{Abs}\;\mathrm{control}\;-\;\mathrm{Abs}\;\mathrm{sample}\right)}{\mathrm{Abs}\;\mathrm{control}\;}\times 100$$


### 2.7. Dye Degradation Study

To study the dye degradation capability of ZnO-NPs, the method reported by Mansi et al. [35] was used. ZnO-NPs were used as probe catalysts to investigate the degradation of methylene blue dye in synthetic wastewater in the presence of sunlight. The experimental methodology includes the dispersion of 100 mg of ZnO-NPs in 200 mL of dye tainted wastewater at a concentration of 10 ppm. To achieve adsorption-desorption equilibrium with dye and ZnO-NPs, the solution was set up in the dark with continuous stirring for 30 min. at 100 rpm. The equilibrated dye and nanoparticles solution was, afterward, subjected to sunlight to study the photocatalysis of ZnO-NPs towards dye degradation. The systematic activity of ZnO-NPs throughout the bulk phase was achieved by continuous stirring at 100 rpm during the process. Thereafter, the reaction mixture was withdrawn after 15 min. intervals for dye concentration evolution. The collected samples were centrifuged at 12,000 rpm to separate the nanoparticles, and finally, the resulting supernatant was subjected to absorption measures using a UV-vis spectrophotometer (Shimadzu corporation, Japan). After a while, the dye absorbance decreased (λ_max_ = 665 nm) gradually, providing the decolorization rate, as well as photocatalytic efficiency of ZnO-NPs, as calculated by using the following equation
η = [(A_0_ − A_t_)/A_0_] × 100
where A_0_ and A_t_ were initial and final absorbance after a certain reaction time, respectively.

### 2.8. Statistical Analysis

All the results were analyzed on MS excel (Microsoft, Redmond, WA, USA), and data were expressed as mean ± SEM. The results were compared with the control group and *p* < 0.05 was considered statistically significant. All the statistical analysis was carried out by SPSS software using paired sample *t*-test.

## 3. Results and Discussion

### 3.1. Characterization of ZnO-NPs

#### 3.1.1. X-ray Diffraction Spectroscopy (XRD)

XRD spectroscopy is a technique used for the determination of as crystalline or amorphous nature of the material. Figure 2 represents the X-ray diffraction pattern of the synthesized ZnO-NPs. The characteristic peaks observed at 31.86°, 34.56°, 36.32°, 47.57°, 56.61°, 62.95°, 66.47°, 68.05°, and 69.16° belongs to (100), (002), (101), (102), (110), (103), (200), (112), and (201) *hkl* planes of ZnO-NPs with hexagonal wurtzite structure [36,37]. A corresponding peak matched well with the standard JCPDS card number: 36-1451 without the presence of any impurity phase. The phase purity was also determined by Rietveld refinement of the given pattern, and structural parameters were obtained as given in Table 1. The Rietveld refinement was performed using FullProf software by describing peak patterns using a pseudo-Voigt profile. Initially, global parameters such as background, scale factors, and cell parameters were refined. Further, the FWHM and atomic orientations were refined consecutively [38]. Figure 2b represents the Rietveld refined patterns of the ZnO-NPs. The broad XRD peaks reveal the lower particle size of ZnO-NPs, and the crystallite size of 20 nm was calculated using the Scherer formula given by [27]:
(1)
$$D=\frac{0.9\lambda }{\beta \cos\theta }$$

where λ is the wavelength (1.54Å), θ is the diffraction angle, and β is the full-width half maximum of the XRD peaks. The lattice strain Ɛ was determined by employing the UDM (Universal Deformation Model) and William Hall equation [39]:
(2)
$$\beta \cos\theta =\frac{K\lambda }{D}+\varepsilon \left(4\sin\theta \right)$$


The UDM model assumes isotropic strain in the crystal in all directions and is independent of the direction of property measurement. The strain generated within the crystal is a result of dislocations and crystal imperfections. The strain was calculated from the slope of the plot between βcos θ along the y-axis and 4sin θ along the x-axis corresponding to each peak of the XRD pattern as given in Figure 2c.

#### 3.1.2. X-ray Photoelectron Spectroscopy (XPS)

The chemical bonding states of the elements of ZnO-NPs were determined using the XPS spectra. Figure 3 shows the O1s and Zn2p core-level XPS spectra of ZnO-NPs. The asymmetric O1s peak was deconvoluted into three different peaks at 531.17 eV, 532.26 eV, and 533.13 eV, which corresponds to the oxygen ions associated with the lattice oxygen of ZnO structure, oxygen vacancies, and chemisorbed oxygen species [40,41]. The chemisorbed species plays a major role in enhancing the antimicrobial and photocatalytic degradation properties [27,39]. In Zn2p core-level XPS spectra shows two symmetrical peaks at 1022.2 eV and 1045.35 eV are ascribed to Zn2p_1/2_ and Zn2p_3/2_. The given value of binding energy is in good agreement with the previously reported data for the Zn^2+^ oxidation state [42].

#### 3.1.3. UV-Visible Spectroscopy 

Figure 4 shows the UV absorbance spectra and Tauc’s plot for the determination of optical bandgap. It can be observed that the absorption spectra of the synthesized ZnO-NPs exhibit a strong absorption band at 353 nm. The excitonic absorption peak can be observed at 257 nm, which is a result of the lower bandgap wavelength of ZnO-NPs than 358 nm [43]. The direct energy bandgap (Eg) that turned out to be 3.21 eV for the synthesized ZnO-NPs was determined using the Tau’s relation [44]:
(3)
$${\left(\alpha h\nu \right)}^{2}=\;A\left(h\nu \;-\;\mathrm{Eg}\right)$$

where α is the absorption coefficient and hν corresponds to the photon energy. Hence, the plot between (αhν)^2^ and (hν) gives the energy bandgap (E_g_).

#### 3.1.4. FE-SEM and Elemental Mapping

The FE-SEM images given in Figure 5a reveal the self-assembly of spherical particles to form a flower-like structure and are arranged uniformly. The agglomeration of larger assembled structures can be observed. Figure 5b shows the elemental distribution of ZnO-NPs, which reveals the uniform distribution of zinc and oxygen in ZnO-NPs. The surface area of nanoparticles is much higher than their bulk counterparts, which allows them to have more molecules on the surface of the nanoparticles. This provides the nanoparticles with their special properties due to their surface chemistry as a photocatalyst and an antimicrobial agent. The plant extract could have played a relevant role as a capping agent and prevented the agglomeration of nanoparticles [39,45].

#### 3.1.5. Transmission Electron Microscopy (TEM) Study

The TEM images reveal the formation of self-assembled grains with irregular shapes and size. It can be observed that the smaller grains diffused to form larger grains. The particle size distribution given in Figure 6b with average particle size of 19.12 ± 0.77 nm was obtained using ImageJ software and origin software. The d-spacing of 0.5906 nm was calculated using ImageJ software from the observed lattice spacing corresponding to (002) *hkl* plane (Figure 6d). The SAED pattern showed the presence of circular rings with bright spots which correspond to the polycrystalline nature of the ZnO-NPs (Figure 6).

#### 3.1.6. FTIR

The Fourier transform infrared spectroscopy (FTIR) is carried out to study the chemical bonds in a molecule by generating the infrared transmittance spectrum. The molecular fingerprint design was used to investigate the different compounds present in the sample. The fingerprint region in the range of 1500–500 cm^−1^ is distinctive for every sample and the region in the range between 4000 to 1500 cm^−1^ is the functional group region. Figure 7 represents the FTIR spectrum of fruit extract and green-synthesized ZnO-NPs to identify the functional groups of phytochemical molecules. The synthesized ZnO-NPs exhibited strong bands at 3432, 2926, corresponding to O–H, N–H stretching vibrations of carboxylic acids, alcohols, amide, respectively. The absorption peaks at 1626 and 1395 cm^−1^ owing to the –C=C– and C–N stretching vibrations of aromatics and aromatic amines [46]. The absorption bands at 898 cm^−1^ and 563 cm^−1^ confirmed the formation of metal oxide [47]. The strong band stretching absorption peaks around 3200–3300 cm^−1^ was assigned to strong band interrelated overlapping stretching vibration of the amide (N–H) and hydroxyl group (OH) in the fruit extract. The OH group stretching indicated the presence of alcohol and phenol and also probably includes the N-H group of proteins [15,48,49]. The fruit extract showed the presence of phytochemical compounds (phenols, glycosides, tannins, saponins, and flavonoids) in *R. ellipticus* [25,26] that facilitate the formation of ZnO-NPs by acting as a capping, reducing, and stabilizing agent [15,27]. The analysis of peaks was in accordance with an earlier study of green synthesis of ZnO-NPs using different extracts [27,46,47].

### 3.2. Antioxidant Activity 

The results of antioxidant potential of fruit extract and ZnO-NPs analyzed through DPPH, ABTS, and FRAP assays are presented in Appendix A and Figure 8. The results indicated that as the concentration of fruit extract, ZnO-NPs, and ascorbic acid increased from 20–100 µg/mL, the percentage inhibition was also increased (Appendix A). In DPPH assay, the IC_50_ values for fruit extract, ZnO-NPs, and ascorbic acid were observed as 32.8 ± 0.5 µg/mL, 72.9 ± 0.7 µg/mL, and 19.5 µg/mL, respectively, whereas for FRAP assays, values were 73.1 ± 0.5 µM Fe^2+^ equivalents, 149.4 ± 0.9 µM Fe^2+^ equivalent, and 45.4 ± 2.4 µM Fe^2+^ equivalents for fruit extract, ZnO-NPs, and ascorbic acid, respectively. In the case of ABTS assay, the IC_50_ values for fruit extract, Ru-ZnO-NPs, and ascorbic acid were observed as 39.2 ± 1.1 µg/mL, 87.48 ± 0.3 µg/mL, and 27.73 ± 1.494 µg/mL, respectively. The paired sample *t*-test revealed a significant variation (*p* > 0.05) between the fruit extract and ZnO-NPs results.

The lower value of IC_50_ indicates a higher antioxidant potential [50]. In DPPH, ABTS, and FRAP assays, ascorbic acid showed the lower IC_50_ followed by fruit extract and ZnO-NPs (Figure 8). This means fruit extract has higher antioxidant potential than ZnO-NPs which could be due to the presence of higher phenolic components in the crude fruit extract of *R. ellipticus*. Mahendran and Kumari [51] and Reddy et al. [52] also observed similar observations in the fruit extract of *Nothapodytes nimmoniana* and *Piper longum* where fruit extracts were observed with higher antioxidant potential than the green-synthesized Ag-NPs. Previous studies also discussed the antioxidant potential of ZnO-NPs synthesized from the plant parts of *Azadirachta indica* [49], *Carica papaya* [53], and *Luffa acutangula* [54]. In all these studies, ZnO-NPs showed a broad spectrum of antioxidant activity by effectively inhibiting the reactive oxygen species.

Nanoparticles exposure causes oxidative stress, which raises concerns regarding their application in humans. However, the green synthesized nanoparticles of this study showed antioxidant activity and might be used to combat oxidative stress. The phytoconstituents of *R. ellipticus* fabricated on the surface of nanoparticles are responsible for this action. The hydroxyl group proved the presence of phenols that are supposed to be attached to the surface of nanoparticles based on FTIR results. These can be used as a substitute for nanoparticles synthesized using other methods. 

### 3.3. Antimicrobial Activity 

The antibacterial activity of *R. ellipticus* fruit extract and ZnO-NPs were tested against Gram-positive (*S. aureus* and *B. subtilis*) and Gram-negative (*E. coli* and *P. aeruginosa*) bacteria using the disc diffusion method and MIC value. Among all bacteria, the maximum inhibition zone was observed against *B. subtilis* (16 ± 0.5 mm) followed by *S. aureus* (14 ± 0.5 mm) and minimum in *P. aeruginosa* (12 ± 0.5 mm) (Figure 9 and Appendix A). Whereas, in the case of fruits aqueous extract, the maximum inhibition zone was achieved for *B. subtilis* (10 ± 1 mm) and lowest for *E. coli* (8 ± 0.5 mm). The results of the disc diffusion method showed higher efficacy of synthesized ZnO-NPs against Gram-positive bacterial cells as compared to Gram-negative bacterial cells. The paired sample *t*-test revealed a significant variation (*p* > 0.05) between the fruit extract and ZnO-NPs results. These results were similar to previous reports on the antibacterial effect of ZnO-NPs [6,49,55,56].

The MIC values of ZnO-NPs against Gram-positive and Gram-negative bacteria ranged from 31.2 to 125 µg/mL, as shown in Table 2. The lowest MIC value was observed against *B. subtilis* (31.2 µg/mL) therefore, indicating the higher effectiveness of the nanoparticles to this microorganism. Whereas for *E. coli* and *S. aureus* MIC value was found to be 62.5 µg/mL and for *P. aeruginosa* MIC value was 125 µg/mL (Table 2). Literature also shows the lower MIC value of ZnO-NPs for *B. subtilis* as compared to other bacteria [57,58]. Whereas Alekish et al. [59] observed the lower MIC value for *S. aureus* as compared to *E. coli*. Ezealisiji and Siwe–Noundou [60] observed the MIC concentration of ZnO-NPs against bacterial pathogens in the range of 0.140 to 6.420 µg/mL.

The antifungal activity of fruit aqueous extract and ZnO-NPs was determined by the food poison technique against two fungal strains (*F. oxysporum* and *R. necatrix*) and further confirmed with MIC value. As shown in Figure 10, ZnO-NPs showed a significantly (*p* > 0.05) higher percent of inhibition of *R. necatrix* (70 ± 1.4%) and *F. oxysporum* (47 ± 1.6%) compared to the fruits aqueous extract (40.8 ± 1.4% and 35 ± 1.4%, respectively) (Figure 10 and Appendix A). Chauhan et al. [27] observed 47% and 26% percentage inhibition of ZnO-NPs for *F. oxysporum* and *R. necatrix*, respectively. On the other hand, Yehia and Ahmed [61] in their study also observed good antifungal activity of ZnO-NPs against *F. oxysporum* and *Penicillium expansum*.

The results of MIC value also showed higher antifungal activity of the ZnO-NPs as compared to fruit’s aqueous extract. Among these two fungi, the ZnO-NPs showed a minimum MIC value against *R. necatrix* (15.62 µg/mL) as compared to *F. oxysporum* (62.5 µg/mL) (Table 2). According to Rosa–Garcia et al. [62], the ZnO-NPs can be used as a strong antifungal agent against plant pathogens.

According to Mahamuni et al. [63], the antimicrobial activity of the ZnO-NPs is due to the release of zinc ions (Zn^+^) in aerobic conditions, which shows toxicity to the microorganism. The released Zn^+^ ions bind to the cell wall and result in disruption of the cell membrane to the production of reactive oxygen species (ROS), including hydrogen peroxide (H_2_O_2_) [63,64]. The generation of hydrogen peroxide is the main factor behind the antimicrobial property. The membrane lipid peroxidation damages the DNA of the cell and also inhibits the plasmid DNA replication. The revealed hydrogen peroxides also disrupt protein structure and reducing sugars, and therefore reduce overall cell viability [65,66,67]. The probable mechanism of antimicrobial activity of ZnO-NPs is presented in Figure 11.

### 3.4. Anticancer Activity 

The anticancer effect of ZnO-NPs on A549 cells was done by MTT assay at different concentrations (1.56–200 µg/mL). The anticancer activity of ZnO-NPs on the A549 cell line revealed that the ZnO-NPs exhibited potent inhibitory activities comparable to that of Paclitaxel, as shown in Figure 12 and Appendix A. ZnO-NPs showed a high percentage (52.41%) of A549 cells death in 200 µg/mL concentration, whereas the IC_50_ value of ZnO-NPs on A549 cells was observed to be 158.1 ± 1.14 µg/mL. Results showed that with increasing concentration (1.56–200 μg/mL) of ZnO-NPs, there was an increase in the death rate of cancer cells. Rajeswaran et al. [68] also observed the potential anticancer activity (IC_50_ 120 µg/mL) of ZnO-NPs synthesized from *Cymodocea serrulata* extract against A549 cell lines. On the other hand, Selim et al. [12] also observed remarkable anticancer activity of ZnO-NPs of aqueous extract of *Deverra tortuosa* on human lung adenocarcinoma (A549) cells. According to Rasmussen et al. [69], the possible mechanism of the anticancer activity of ZnO-NPs is their semiconductor nature, which induces oxidative stress in cancer cells by generation of ROS and therefore leads to the death of cancer cells [69,70,71,72].

### 3.5. Photocatalytic Effectiveness

Methylene blue is a highly toxic cationic dye that is used to dye paper, leather, and textiles. The industrial effluent mixes this dye with river and other water sources, therefore preventing the penetration of solar radiation which eventually affects the water-based photosynthesis, resulting in harm to water ecology, causing environmental pollution, and poisoning the food chain [73]. In Figure 13, the decrease in peak absorbance of methylene blue at 665 nm as a function of irradiation time depicts the photocatalytic efficiency of applied ZnO-NPs, as synthesized ZnO-NPs utilizing *Rubus ellipticus* fruit extract has shown remarkable photodegradation activity against methylene blue contaminated water samples at various degradation intervals. Within the first 15 min. of photo degradation, there was a considerable decline in absorption intensity, which eventually disappeared completely at the end of 180 min. Figure 13b presents the time profile of MB dye degradation efficiency. Within 60 min. of photocatalysis, approximately 17.5% of the total methylene blue concentration was degraded, yielding a final MB degradation efficiency of 72.7% within 180 min. In the early stage of photodegradation, the availability of numerous catalytic sites and a high concentration gradient encouraged the fast colour deterioration [74]. The photocatalytic degradation of methylene blue confirms the amphoteric nature of ZnO-NPs.

In the presence of sunlight, the photodegradation ability of ZnO-NPs proposed a three-step mechanism that involves the generation and transfer of electron-hole pairs, radical generation, and then degradation of dye. The electron and hole pair from ZnO conduction and valance bands are generated in the first step by irradiating the material with sunlight, as explained in Equation (4).
ZnO + hν → e^−^_cb_ + h^+^_vb_(4)

In the second phase, oxygen molecules interact with surface electrons to generate superoxides, which are then converted to peroxide molecules. In addition, the surface holes oxidize the water molecules, resulting in the formation of hydroxyl ions as shown in Equations (5)–(7) [75].
O_2_ + e^−^ → O^°^_2_ + H^+^ → HOO(5)
HOO^°^ + e^−^ + H^+^ → H_2_O_2_(6)
H_2_O + h^+^ → OH^°^ + H^+^(7)

The above-formed intermediates are exceedingly unstable, and when they react with the dye substituent, they lead to disintegration into mineralized products, as presented in Equation (8) [76].
MB Dye molecule + (OH^°^, HOO^°^, H^+^ or H_2_O_2_) → Degraded/mineralized products.(8)

A large surface area was provided by the spherical shape of green synthesized ZnO-NPs to facilitate electron-hole pair separation for distinct radical species that readily helps to disintegrate the dye molecule. However, the dye degradation efficiency of ZnO-NPs is dramatically lowered due to severe aggregation. The findings of photodegradation investigations revealed significant photocatalytic activity of the synthesized ZnO-NPs. The almost total disintegration of methylene blue, with around 17.5% photodegradation within 1 h, indicates that as-prepared ZnO-NPs had substantial photocatalytic efficiency.

## 4. Conclusions

Nanotechnology is evolving rapidly, yet research into the toxicological effects of nanoparticles on public health and the environment is still in its infancy. In the present study, the ZnO-NPs were synthesized using fruit extract of *R. ellipticus* for the first time via a simple, low-cost, and environment-friendly method. The fruit’s aqueous extract acts as a capping and stabilizing agent. The X-ray diffraction patterns revealed the formation of the pure phase ZnO-NPs the hexagonal wurtzite structure with a crystallite size of 20 nm. The Rietveld refinement confirmed the phase purity of ZnO-NPs with the p-63-mc space group and a lower chi^2^ value (χ) of 2.15. FTIR spectroscopy was utilized to determine the functional groups associated with the ZnO-NPs and the Z-O stretching band was observed at 536 cm^−1^. XPS revealed the presence of oxygen chemisorbed species on the surface of ZnO-NPs. UV-vis spectroscopy showed the characteristic absorption peak of ZnO-NPs at 353 nm with an optical bandgap of 3.21 eV obtained using Tau’s plot. FE-SEM and elemental mapping revealed the flowerlike morphology of self-assembled grains with a uniform distribution of elements. The TEM and SAED showed the formation of self-assembled grains with irregular shapes, size, and polycrystalline nature of the ZnO-NPs.

ZnO-NPs have shown antioxidant, antibacterial (against Gram-positive and Gram-negative bacteria), antifungal (plant pathogenic fungi), and anticancer activity (cancer cell (A549) lines through in vitro assays. The antioxidant activity exhibited could be due to the attachment of the -OH group of polyphenolic compounds on the surface of ZnO-NPs. The highest antibacterial activity of ZnO-NPs was observed against *B. subtilis* followed by *S. aureus*, and *E. coli* whereas, the maximum antifungal activity was observed against *R. necatrix* then *F. oxysporum*. The anticancer assay showed an increased death rate of cancer cells with increasing concentration (1.56–200 μg/mL) of ZnO-NPs. The ZnO-NPs have shown significant photocatalytic activity by degrading nearly 17.5% of methylene blue dye within 1 h.

In conclusion, the green synthesis technique is beneficial for plant-based fabrication. Due to less toxicity and chemical-free nature, and significant biological activities, the ZnO-NPs synthesized from fruits aqueous extract of *R. ellipticus* can be considered as biocompatible nanomaterials. Hence, these ZnO-NPs are safe and can be used for wastewater treatment and for medicinal purposes by pharmacological industries. From a future perspective, the in vivo study is required for their biological activities and phytoconstituents present in plants will develop a novel platform for green synthesis of nanoparticles for their biomedical purpose.

## Figures and Tables

**Figure 1 materials-15-03470-f001:**
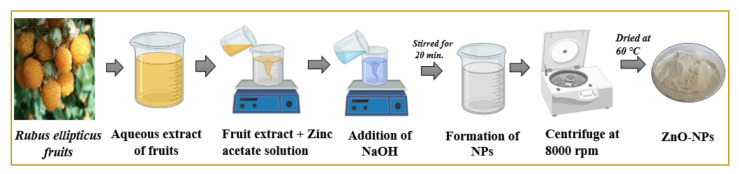
Schematic representation of biosynthesis of ZnO-NPs using fruit extract.

**Figure 2 materials-15-03470-f002:**
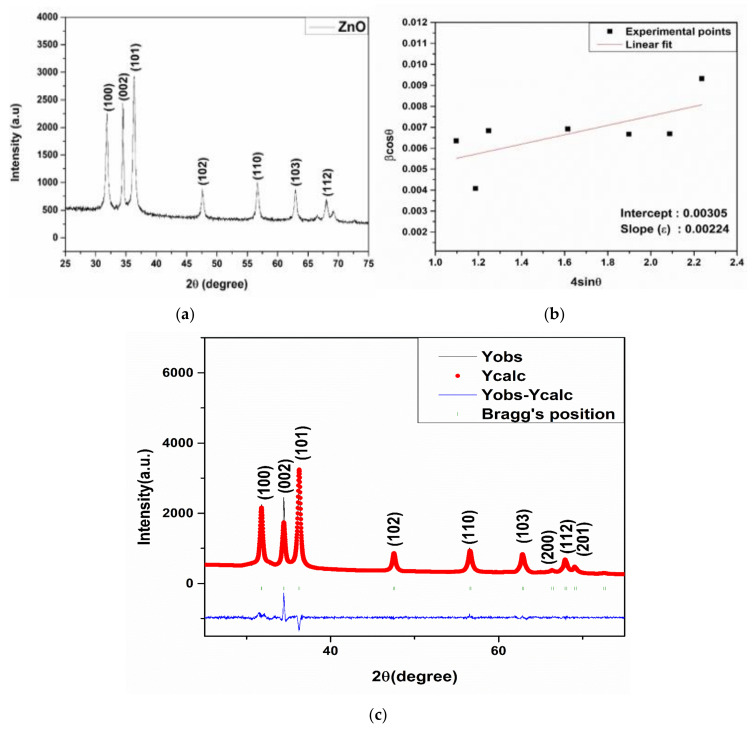
(**a**) XRD pattern, (**b**) W-H plot, and (**c**) Rietveld refined pattern of ZnO-NPs.

**Figure 3 materials-15-03470-f003:**
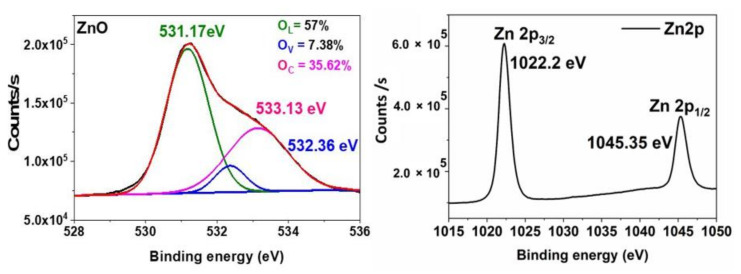
O1s and Zn2p core-shell XPS spectra of ZnO-NPs.

**Figure 4 materials-15-03470-f004:**
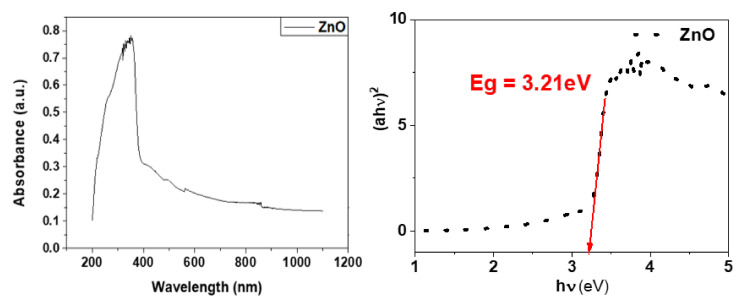
UV-vis Absorbance and Tauc’s plot of ZnO-NPs.

**Figure 5 materials-15-03470-f005:**
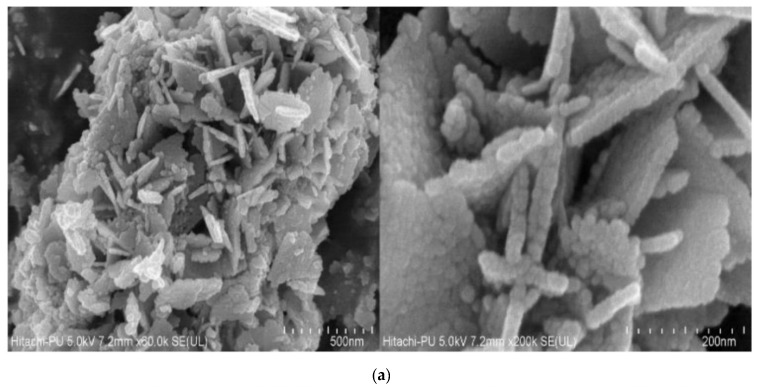
FE-SEM micrographs (**a**) and Elemental mapping (**b**) of ZnO-NPs.

**Figure 6 materials-15-03470-f006:**
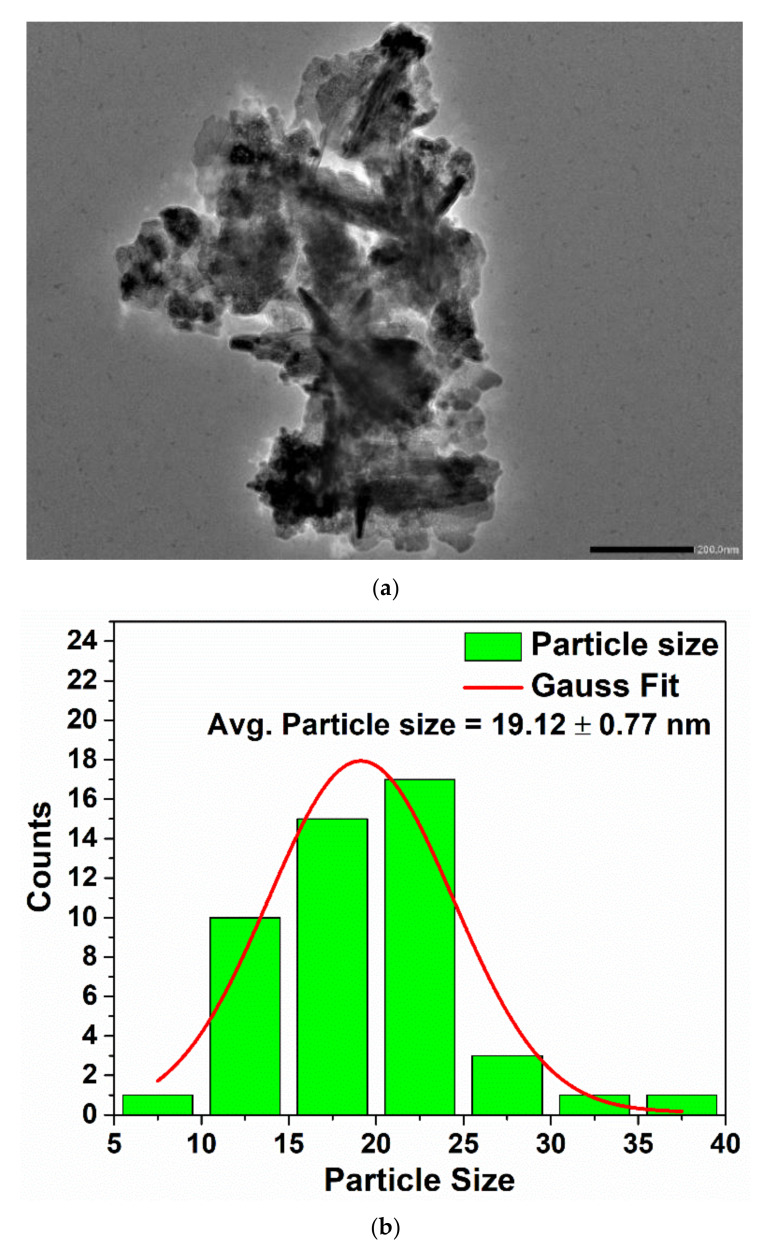
TEM micrograph (**a**), particle size distribution (**b**), SAED pattern (**c**) and d—spacing (given at 20 nm) (**d**) of ZnO-NPs.

**Figure 7 materials-15-03470-f007:**
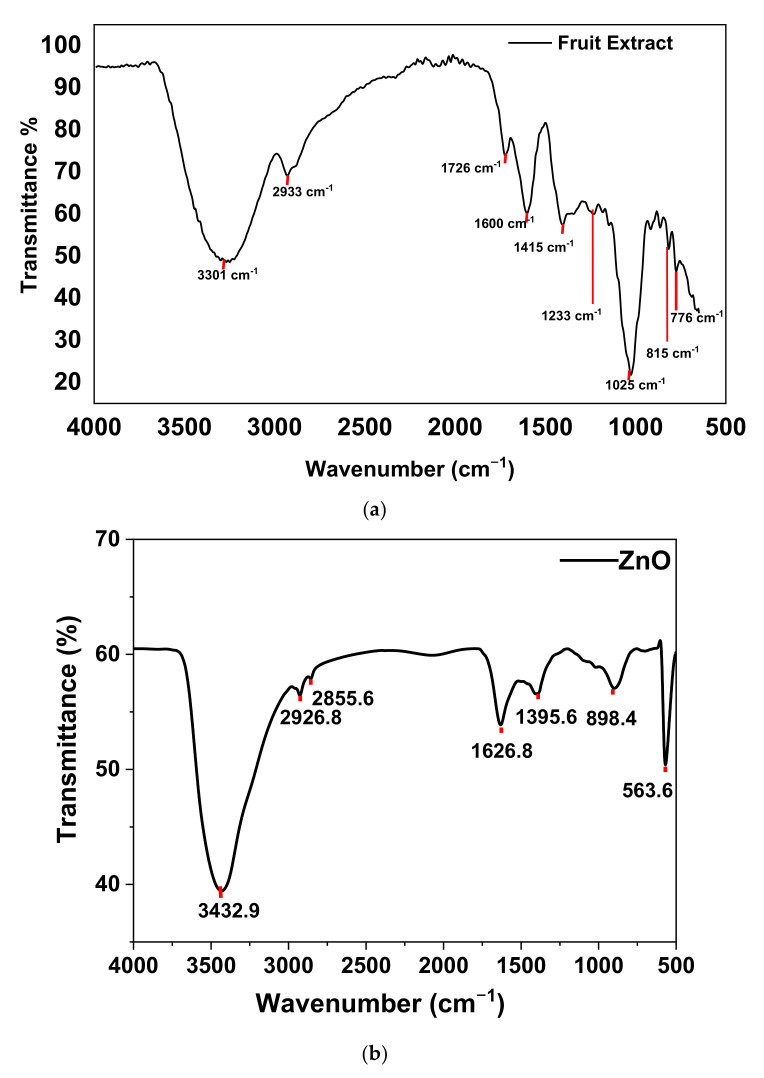
FTIR of (**a**) fruit extract and (**b**) ZnO-NPs.

**Figure 8 materials-15-03470-f008:**
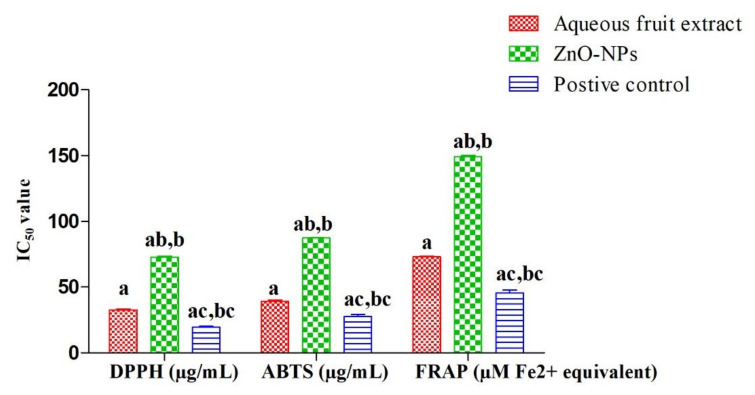
In vitro antioxidant potential (IC_50_ value) of zinc oxide nanoparticles and fruit extract [Different superscript showed significant (*p* < 0.05) difference between (a) fruit extract, (b) ZnO-NPs and (c) positive control between plant extract and nanoparticles].

**Figure 9 materials-15-03470-f009:**
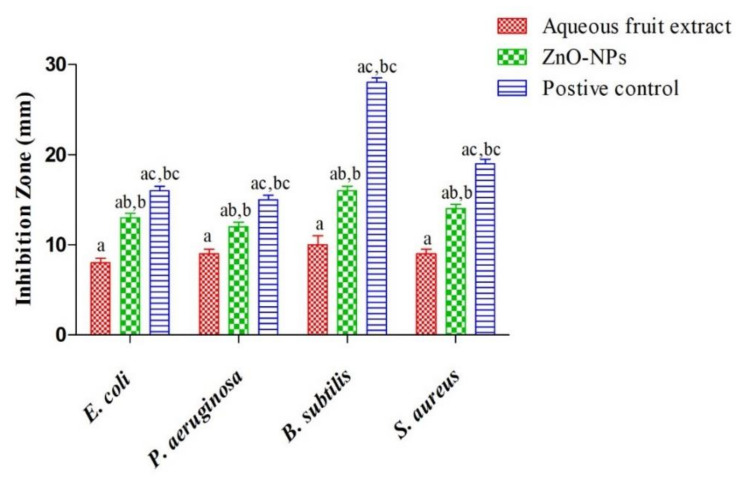
Antibacterial activity of aqueous extract and ZnO-NPs against pathogenic bacteria [Different superscript showed significant (*p* < 0.05) difference between (a) fruit extract, (b) ZnO-NPs, and (c) positive control].

**Figure 10 materials-15-03470-f010:**
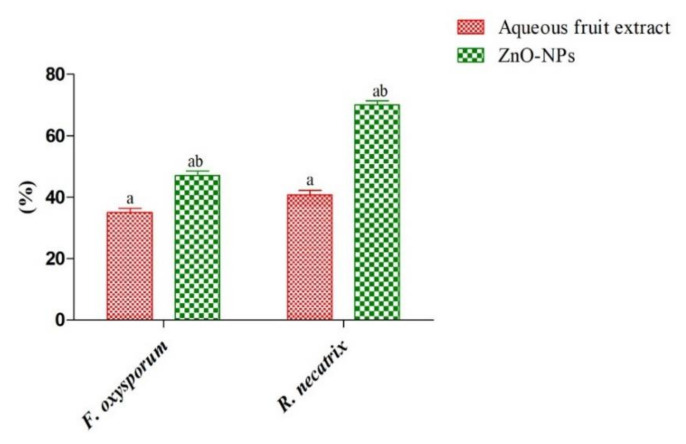
Graphical representation of the antifungal activity of plant extract and ZnO-NPs against plant pathogenic fungi [Different superscript showed significant (*p* < 0.05) difference between (a) fruit extract, (b) ZnO-NPs].

**Figure 11 materials-15-03470-f011:**
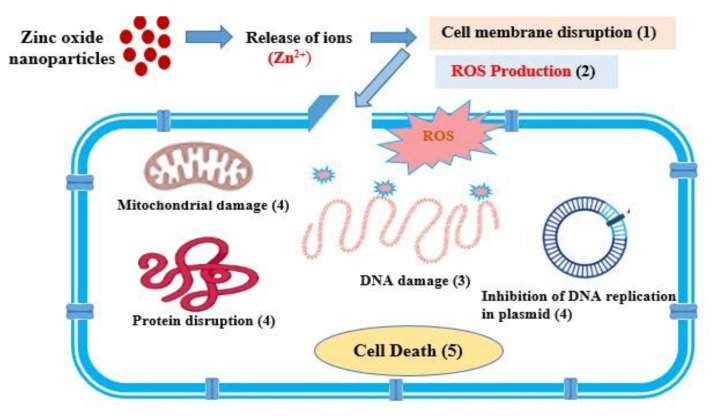
Antimicrobial mechanism of ZnO-NPs.

**Figure 12 materials-15-03470-f012:**
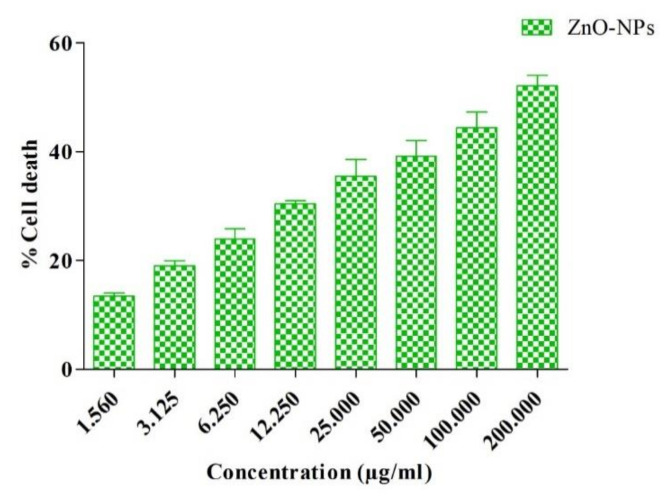
Anticancer effect of ZnO-NPs at a concentration (1.56–200 µg/mL) on A549 cells.

**Figure 13 materials-15-03470-f013:**
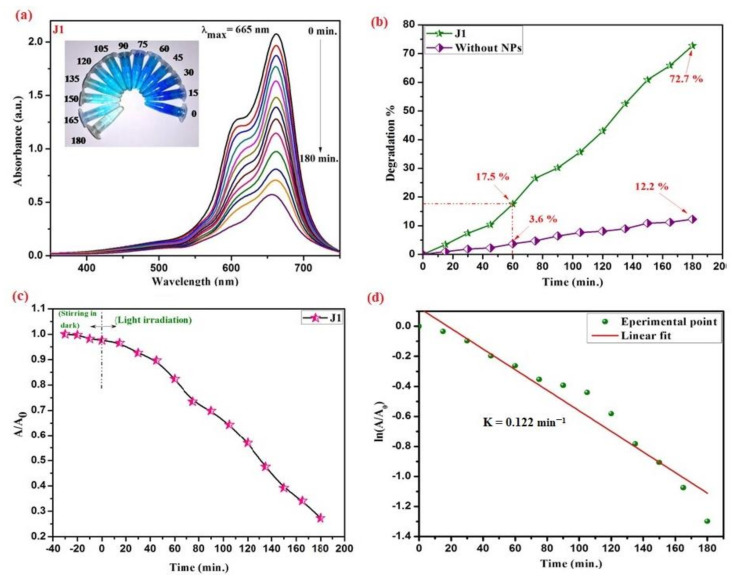
Photocatalytic dye degradation studies by ZnO-NPs (**a**) Absorption spectrum of MB dye samples along with decoloration of dye from dark blue to a nearly clear solution, as shown in the inset; (**b**) Time profile of degradation efficiency for MB dye (**c**) Normalized absorbance of dye in dark and sunlight in the presence of a catalyst (**d**) Kinetic study and rate constant calculated for the dye degradation reaction.

**Table 1 materials-15-03470-t001:** Structural and Rietveld refine parameters of ZnO-NPs.

Sample	ZnO-NPs
Structure	Hexagonal-wurtize
Space group	P-63-mc
a (Å)	3.251287
b (Å)	3.251287
c (Å)	5.208760
Volume (Å)^3^	47.684
$\chi^{2}$	2.15
$R_{p}$	14.7
$R_{wp}$	21.5
$R_{e}$	25.4
Average Crystallite size (Scherer method) (nm)	20
Strain (Ɛ)	0.00224

**Table 2 materials-15-03470-t002:** MIC of fruit extract and ZnO-NPs against bacterial and fungal strains.

Microbial Strains		MIC (µg/mL)	
	Fruit Extract	ZnO-NPs	Ampicillin
*E. coli*	250	62.5	6.25
*P. aeruginosa*	250	125	6.25
*B. subtilis*	500	31.2	1.56
*S. aureus*	500	62.5	3.12
*F. oxysporum*	125	62.5	–
*R. necatrix*	62.5	15.6	–

## Data Availability

All data are available in the manuscript.

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
