# Peer review of "Rubus ellipticus Sm. Fruit Extract Mediated Zinc Oxide Nanoparticles: A Green Approach for Dye Degradation and Biomedical Applications"

_materials, 2022, doi:10.3390/ma15103470_

Round 1

Reviewer 1 Report

Dear Authors, in your interesting manuscript, the following points should be added/changed to further improve it:

  1. Introduction: I have a comment on the sentence “Different types of metallic nanoparticles have been synthesized through this technique like silver oxide (AgO), silicon dioxide… (45-46). I note that the authors give the example of metal oxides NPs and not matals NPs.
  2. Introduction: I have a comment on the sentence “The Zinc oxide nanoparticles (ZnO-NPs) are easy to synthesize, showed less toxicity, therefore, has been reported as the most popular metal oxide nanoparticle. (49-51)” I ask the authors to provide references (where this was reported).
  3. Introduction: I have a comment on the sentence The various methods that have been used for the synthesis of ZnO-NPs are the solgel method [7], chemical [8], hybrid [9], direct oxidation [10], and biological or green synthesis methods [11,12] (55-57)“. Please explain to me what the article on "Raw clay supported ZnO nanoparticles in photodegradation of 2-chlorophenol under direct solar radiations" has to do with the preparation of ZnO by direct oxidation. I suggest the authors to add as references significant review articles on the synthesis of ZnO (DOI:10.3390/nano10061086, DOI:10.1080/10408436.2021.1886041, DOI:10.1016/j.inoche.2020.108140). One of the important methods for the synthesis of nano ZnO is the microwave method, which is worth mentioning in the mentioned sentence as an example.
  4. Synthesis of ZnO-NPs: Please specify the concentration of fruits aqueous extract used. Please add information on zinc acetate (purity, manufacturer).
  5. Synthesis of ZnO-NPs: The authors state that the method described by Chauhan et al. [27] was used for the synthesis of ZnO NPs. So please explain me why the authors did not mention the use of sodium hydroxide in the synthesis of ZnO NPs?
  6. Characterization of ZnO-NPs: The authors mentioned in the abstract that they also used the TEM research method. A description of this method is missing.
  7. Results and Discussion: Please explain to me why the authors did not obtain a ZnO sample without the addition of plant extract as a reference sample to compare the effect of the samples (Does a green approach matter?).
  8. FE-SEM and Elemental Mapping: I have a comment on the sentence “Fig. 5(b) shows the elemental distribution of ZnO NPs, which reveals the uniform distribution of zinc and oxygen in ZnO-NPs.” Please explain to me how it was possible to conclude from scale bar EDS results of 100nm size that in ZnO NPs of about 30nm size the distribution of zinc and oxygen is homogeneous? I note that the sample for EDS mapping should be smooth.
  9. Conclusion: I have a comment on the sentence “The Rietveld refinement confirmed the phase purity of ZnO-NPs with the p-63-mc 508 space group and a lower chi2 value (χ) of 2.46.” Please indicate to me in the manuscript where the chi2 value (χ) of 2.46 was calculated?
  10. Conclusion: I have a comment on the sentence “ZnO-NPs the hexagonal wurtzite structure with a crystallite size of 20nm.” The abstract reports that the average crystallite size is 22 nm.

Author Response

The authors want to thank the reviewer for the valuable input in the review of this article.

Please see the attached document with the comments of the reviewers addressed. 

Reviewer 2 Report

In this manuscript, the authors synthesized ZnO nanoparticles using fruit extract as the reducing and capping agent. The structure and chemical properties of the as-received material were well-characterised. The authors have tested these materials for a variety of applications, which have shown promising activities. In general, the manuscript is well-structured and fits the scope of the journal. However, the author should provide more information on how the ZnO affects the performance. Therefore, I would recommend the manuscript to be published in Materials after the authors address the below-listed comments:

  1. In the dye degradation measurement, the author should specify and control the light intensity of the sunlight, in order to get reproducible results.
  2. The FTIR results showed that the ZnO also has carboxylic acids, alcohols, and amides on the surface. Are they from the chemisorbed fruit extract? It will be beneficial if the authors can elaborate more on the nature of these surface species.
  3. Will the surface species affect the tests?
  4. The authors claimed that “The rate of recombination of charge carriers is decreases with sunlight irradiation and facilitating electron hole pair separation on the surface.” This is incorrect. The rate of recommendation of charge carriers and separation are intrinsic properties of the semiconductor materials. Light irradiation can only induce such processes but not inhabit/facilitate them.
  5. The authors should check the grammar throughout the manuscript. There are some incomplete sentences, for example: “The findings of photodegradation investigations revealed that as-synthesized ZnO-NPs significant photocatalytic activity.”

Author Response

(The authors gave the same response as above.)

Round 2

Reviewer 1 Report

Comment 1: FE-SEM and Elemental Mapping: I have a comment on the sentence “Fig. 5(b) shows the elemental distribution of ZnO NPs, which reveals the uniform distribution of zinc and oxygen in ZnO-NPs.” Please explain to me how it was possible to conclude from scale bar EDS results of 100nm size that in ZnO NPs of about 30nm size the distribution of zinc and oxygen is homogeneous? I note that the sample for EDS mapping should be smooth.

Reply: We appreciate the reviewers concern but 20 nm is the crystallite size however which we observed in SEM image are the grains formed by the accumulation of the crystallites. In the mapping images, a fluorescent colour zone denotes a high content of comparable elements that are uniformly distributed across all particles. The even colour distribution of Zn and O indicates that ZnO nanoparticles were successfully prepared.

Given below are few references supporting our statement: -

  1. Khezami L, Modwi A, Ghiloufi I, Taha KK, Bououdina M, ElJery A, El Mir L. Effect of aluminum loading on structural and morphological characteristics of ZnO nanoparticles for heavy metal ion elimination. Environmental Science and Pollution Research. 2020 Jan;27(3):3086-99.
  2. Pascariu P, Tudose IV, Suchea M, Koudoumas E, Fifere N, Airinei A. Preparation and characterization of Ni, Co doped ZnO nanoparticles for photocatalytic applications. Applied surface science. 2018 Aug 1;448:481-8.

Reviewer's reply: The authors concluded that „Fig. 5(b) shows the elemental distribution of ZnO NPs, which reveals the uniform distribution of zinc and oxygen in ZnO-NPs.” Firstly, I would ask the authors to determine/ state the average size of the ZnO NPs obtained since they claim "which reveals the uniform distribution of zinc and oxygen in ZnO-NPs" and “In the mapping images, a fluorescent colour zone denotes a high content of comparable elements that are uniformly distributed across all particles”. Secondly, please explain to me why in the image of Fig. 5b (SE, Zn and O) the whole area of visible ZnO NPs is not marked in red and green? I note that EDX analysis has many limitations that authors forget or are unaware of. I suggest not to duplicate the mistakes of other authors when discussing EDS result. Third, the authors state "The even colour distribution of Zn and O indicates that ZnO nanoparticles were successfully prepared." I note that the evidence confirming that zinc oxide was obtained is the XRD result (Fig. 2 (a)). However, SEM/TEM results should confirm that spherical nano zinc oxide (particles) have been obtained. I note again that despite using two SEM and TEM methods, the authors did not determine/calculate the size of the nanoparticles obtained. Does this mean that the reader has to guess (draw conclusions) whether nano has been obtained? Please consider my suggestions and improve the discussion of the results. I note that the authors have added a sensational TEM result, a beautiful image showing the size of the spherical ZnO particle, but the scale bar is unreadable (Fig. 6. (b)). 

Comment 2: Please improve the readability of the scale bar on TEM images (Fig. 6).

Comment 3: Please name each image in the title description ((a)....,(b).....,(c)......) (Fig. 5 and Fig. 6).

Good Luck!

Author Response

The authors want to thank the reviewer for the additional comments to improve the article. The comments are addressed in the attached file. 

Kind regards
